# Relationship between Tissue Gliding of the Lateral Thigh and Gait Parameters after Trochanteric Fractures

**DOI:** 10.3390/s22103842

**Published:** 2022-05-19

**Authors:** Kengo Kawanishi, Daisuke Fukuda, Hiroyuki Niwa, Taisuke Okuno, Toshinori Miyashita, Takashi Kitagawa, Shintarou Kudo

**Affiliations:** 1Graduate School of Health Sciences, Morinomiya University of Medical Sciences, Osaka 559-8611, Japan; riverwest1004@gmail.com (K.K.); takashi.k.0083@gmail.com (T.K.); 2Department of Rehabilitation, Kano General Hospital, Osaka 531-0041, Japan; liberte_rhythm2o@icloud.com (H.N.); taisuke.okuno24@gmail.com (T.O.); 3Inclusive Medical Sciences Research Institute, Morinomiya University of Medical Sciences, Osaka 559-8611, Japan; miyashita.osaka@gmail.com; 4Department of Physical Therapy, Morinomiya University of Medical Sciences, Osaka 559-8611, Japan; reportmatchtrick@gmail.com; 5Department of Rehabilitation, Higashi Osaka Hospital, Osaka 536-0005, Japan; 6PMP Inc., Osaka 580-0025, Japan

**Keywords:** gliding, jerk, trochanteric fracture

## Abstract

Trochanteric fractures lead to severe functional deficits and gait disorders compared to femoral neck fractures. This study aims to investigate gait parameters related to gliding between tissues (gliding) after trochanteric fracture (TF) surgery. This study implemented a cross-sectional design and was conducted amongst patients who underwent TF surgery (*n* = 94) approximately three weeks post-trochanteric fracture surgery. The following parameters were evaluated: (1) gliding between tissues; (2) lateral femoral pain during loading; (3) maximum gait speed; (4) stride time variability and step time asymmetry as measures of gait cycle variability; (5) double stance ratio and single stance ratio for assessment of stance phase, (6) jerk; and (7) Locomotor rehabilitation index as a measure of force changes during gait. The gliding coefficient was significantly correlated with lateral femoral pain (r = 0.517), jerk root mean square (r = −0.433), and initial contact-loading response jerk (r = −0.459). The jerk of the force change value during gait was also effective in understanding the characteristics of the gait in the initial contact-loading response in patients with trochanteric fractures. Additionally, gliding is related not only to impairments such as pain but also to disabilities such as those affecting gait.

## 1. Introduction

The global incidence of hip fractures is estimated to increase by 6.3 million every year until 2050 [1]. Postoperative orthopedic complications include dysfunctions such as abnormal gait [2]. In particular, trochanteric fractures lead to severe functional deficits and gait disorders compared to femoral neck fractures [3,4,5,6]. Severe functional deficits and gait disorders after trochanteric fractures indicate the influence of different fracture patterns and surgical approaches [5]. Therefore, trochanteric fractures are often difficult to treat in rehabilitation medicine. In recent years, it has been reported that gliding between the subcutaneous tissue and vastus lateralis (VL) in the lateral thigh is important for lateral femoral pain during loading after trochanteric fracture surgery [7]. Gliding between tissues is also thought to be related to gait disorders following trochanteric fracture surgery. However, the relationship between gait and gliding between tissues is still unclear.

Gait velocity is a simple, highly valid, and reproducible variable and is assessed for some gait disorders in clinical settings [8,9]. However, gait disorders after trochanteric fracture may be related to various factors such as age, sex, fracture type, balance ability, and cognitive function. Therefore, it is unclear whether gait velocity is sufficiently indicative of gait disorders. Patients with trochanteric fracture surgery have lateral femoral pain, and a decrease in gliding between tissues typically shows a gait disorder during the loading response (LR) phase of the gait [7]. However, gait observation and measurement velocity cannot represent gait disorders during the LR phase.

The use of inertial sensors with high portability, validity, and reliability has made it possible to measure gait easily [10]. Inertial sensors not only assess spatiotemporal parameters, but also kinematic and kinetic analyses [11,12,13] during gait in clinical settings.

We hypothesized that gliding between the VL and subcutaneous tissue is associated with gait disorders during the LR phase of gait motion. The purpose of this study was to investigate the relationship between gliding in the VL and subcutaneous tissue and the parameters as measured by inertial sensors.

## 2. Materials and Methods

### 2.1. Participants

Ninety-four patients with TF participated in this study from two hospitals (male: *n* = 19, female: *n* = 75, mean age: 82.9 ± 6.7 years). The patients underwent open reduction and internal fixation (gamma nails). A survey was conducted at approximately three postoperative weeks. The study duration was from June 2019 to December 2020, and the inclusion criterion was pain in the mid portion of the lateral thigh. The exclusion criteria were: (1) patients who had difficulty walking more than 10 m without assistance; (2) patients who could not report their pain accurately due to severe dementia; (3) patients with hemiplegia due to stroke; (4) patients whose gait speed was more than 1 m/s; and (5) patients who could not provide consent to participate in the study. 

### 2.2. Ethical Statements

The study was conducted in accordance with the Declaration of Helsinki, and the protocol was approved by the ethics committee of the Morinomiya University of Medical Sciences (approval number: 2019-087), and all the patients provided written informed consent.

### 2.3. Outcome Measures

The following parameters were evaluated: (1) gliding between tissues; (2) lateral femoral pain during loading; (3) maximum gait speed; (4) stride time variability (STV) [14] and step time asymmetry (STA) [15] as measures of gait cycle variability; (5) double and single stance ratios for the assessment of stance phase [16]; (6) jerk [12,17] as a measure of force changes during gait; and (7) Locomotor rehabilitation index [18].

### 2.4. Procedure of Gait Assessments 

An inertial sensor (MicroStone Corporation, Nagano, Japan, MVP-RF8-HC, sampling frequency, 100 Hz) and the tablet (ASUSTeK Computer Inc., Taipei, Taiwan, Nexus 7; sampling frequency, 30 Hz) were used for gait assessment. The inertial sensor was placed on a flat surface and calibrated. For various gait analyses, such as asymmetry, an inertial sensor was attached to the third lumbar vertebra [15]. The inertial sensor was securely fixed on the lower back (third lumbar vertebra) using an elastic belt which was placed over the clothes of the participant. Subsequently, the patients walked with maximum effort on a 14-m gait path, including a 2-m reserve path in front and behind, and each patient’s gait speed was measured. Additionally, the initial contact (IC) and toe-off of the left and right sides of the gait cycle were identified from the measured acceleration waveform and video image files captured by a tablet linked to the inertial sensor. The initial data of the measured acceleration waveforms were collected on a tablet [19,20]. 

### 2.5. Data Analysis

Initial data was analyzed with stable accelerometer waveform data during steady walking over five steps. Based on IC and toe-off of the left and right sides of the gait cycle, we calculated the percentage of single-leg and double-support phases on the affected side in one gait cycle. The STV [13], STA [14], and jerk [11,16] in phases 6–10 of the gait cycle were also calculated. The STV was calculated as the standard deviation of each stride time divided by the mean stride time [14,18]. The STA was calculated using the difference between the left and right stance times divided by the total number of times of the left and right stances [15]. Raw linear acceleration data in three axes (anterior-posterior, medio-lateral, and superior-inferior axes) were low-pass Butterworth filtered at 2 Hz. The jerk was calculated from the first-time derivative of the linear acceleration. The root mean square (RMS) of each axis was used to calculate the jerk composite vector along the three axes (Equation (1)), respectively.
(1)$$m=\sqrt{x^{2}+y^{2}+z^{2}}$$

This equation determines the immediate magnitude of vectors by calculating the square root of the sum of each axis when squared, which condensed the three-axis value. To quantify the change in force during gait, the average RMS value of the jerk composite vector during one gait cycle was calculated as jerk RMS [13] (Figure 1). Additionally, the peak values of the IC-LR jerk composite vector were measured five times in every 6–10 phases of the gait cycle to determine the change in force in the IC-LR [12] (Figure 1). The mean value was calculated using the IC-LR jerk. The IC-LR was defined as 12% of one gait cycle [20].

The locomotor rehabilitation index is a method that can determine the similarity between self-selected walking speed and optimal walking speed [18]. First, the theoretical optimal walking speed of the elderly was calculated using the mathematical model as in Equation (2) [21].
(2)$$Optimal\;Walking\;Speed\;of\;the\;elderly=\sqrt{0.25\times 9.81\times lower\;limb\;length}$$

Then, the locomotor rehabilitation index is calculated from the ratio of the self-selected walking speed to the theoretical optimal walking speed multiplied by 100, according to Equation (3) [21].
(3)$$Locomotor\;Rehabilitation\;Index=\frac{self-selected\;walking\;speed}{Optimal\;Waking\;Speed\;of\;the\;elderly}\times 100$$

### 2.6. Pain Assessment

Lateral femoral pain during the stance phase of the gait was assessed using the numerical rating scale (NRS).

### 2.7. Evaluation of Gliding between the Subcutaneous Tissue and the Vastus Lateralis

Gliding between tissues was applied according to previous studies [7,22] as follows: Ultrasonography (Canon Aplio 500; Canon Co., Ltd., Tokyo, Japan) and a 12 MHz linear probe (PLT1204ST; Canon Co. Ltd., Tokyo, Japan) were used to capture the dynamics of the lateral thigh during flexion-extension of the knee joint. Subsequently, the gliding coefficient was calculated from the flow velocity in the subcutaneous tissue and superficial VL using the flow vector analysis software (Flow PIV flow vector analysis software; Library Co., Ltd., Tokyo, Japan). A lower gliding coefficient was defined as better gliding between tissues.

### 2.8. Statistical Analysis 

Measured data were compared to normal values using the Kolmogorov-Smirnov test. The Pearson product-moment correlation coefficient and Spearman’s rank correlation coefficient were used to examine the correlation between the gliding coefficient, lateral femoral pain, and gait parameters. All statistical analyses were performed using IBM SPSS Statistics for Windows (IBM SPSS Statistics for Windows, Version 24.0. Armonk, NY, USA). Statistical significance was set at *p* < 0.05. The sample size was estimated for a power of 0.8 and an effect size of 0.6. The sample size was determined by GPower 3 Software (version 3.1.9.4) using the T test for correlation. Eighteen participants were deemed adequate (with one extra to account for possible data attrition).

## 3. Results

Of the 94 participants, 18 participants met the inclusion criteria and were included in the study (male: *n* = 1, female: *n* = 17, mean age: 84.9 ± 7.0 years, Height: 149.4 ± 7.0 cm, lower limb length: 70.0 ± 5.2 cm, and postoperative: 34.1 ± 15.3 days) (Figure 2). Reasons for the exclusion of participants are listed as follows. (1) Patients who did not provide consent for participation (*n* = 7), (2) 15 patients with severe dementia, (3) 17 patients without lateral femoral pain on loading, (4) 24 patients who had difficulty with gait for more than 10 m, (5) 11 patients having hemiplegia due to stroke, and (6) Patients having gait speed more than 1 m/s (*n* = 2). No postoperative problems such as excessive sliding of the lag screw or cut-out were observed in any of the patients.

The results of measurement data are shown in Table 1. The results of the Kolmogorov-Smirnov test confirmed normality, except for the STV, STA, and double stance ratio. The gliding coefficient was significantly correlated with lateral femoral pain (r = 0.517), jerk RMS (r = −0.433), and IC-LR jerk (r = −0.459) (Table 2). The maximum gait velocity was significantly correlated positively with jerk RMS (r = 0.599), IC-LR jerk (r = 0.679), double stance ratio (r = −0.608), single stance ratio (r = 0.531), and Locomotor rehabilitation index (r = 0.998) (Table 3). Additionally, the jerk RMS showed a significant positive correlation with the IC-LR jerk (r = 0.940) and Locomotor rehabilitation index (r = 0.742). Participants with decreased gliding between tissues indicated lower jerk RMS and IC-LR jerk (Figure 3).

## 4. Discussion

The purpose of this study was to clarify the relationship between the gliding of the lateral thigh following trochanteric fractures and gait parameters. The results showed that gliding had a moderate negative correlation with the jerk RMS and IC-LR jerk. The jerk RMS is strongly correlated with IC-LR jerk. In other words, the change in the force applied to the trunk in the LR phase is considered to be small in cases with low gliding. Additionally, jerk is a parameter that is related to gait velocity but is not associated with lateral femoral pain. Therefore, gliding is related to lateral femoral pain and jerk during gait but not directly to lateral femoral pain and gait. Moreover, gliding was found to be an important parameter that could affect lateral femoral pain and gait velocity. In other words, the jerk of the trunk during gait was low when gliding decreased.

This study did not find any postoperative problems such as excessive sliding or cut-out of the lag screw. Therefore, biomechanical parameters such as leg length difference that affect gait were not affected. In our previous study, we reported that lateral femoral pain after trochanteric fracture was associated with gliding [7] and that this gliding was related to subcutaneous tissue thickness and dense connective tissue ratio [22]. However, the relationship between gliding and gait is unclear. The conventional gait assessment for trochanteric and femoral neck fractures uses the spatiotemporal parameters [16] and analysis of dynamic weight loading during the stance phase [5] to characterize gait. However, all previously reported studies are limited to the analysis of variability throughout the gait cycle or analysis of the dynamic weight load during the stance phase. After the trochanteric fractures, the lateral femoral pain is frequent [7], and the large ground reaction force in the IC-LR phases [23] presents challenges. However, there is an assessment of kinetics in the subdivision of the stance phase. Therefore, we analyzed the jerk based on the values of the inertial sensors fixed on the trunk. Previous studies of gait analysis using jerk have been used for knee osteoarthritis [12,13,14,15,16,17,24] and Parkinson’s disease [13]. The results of this study showed that patients with trochanteric fractures had a significant decrease in gait speed and a small change in force in the IC-LR. Additionally, there was no correlation between the conventional gait parameters and the gliding coefficient. Therefore, we analyzed the jerk, which indicates the change in force and added it to the gait parameters. We were then able to clarify the relationship of gliding between tissues and gait after trochanteric fractures.

In previous studies, jerk was considered an index to quantitatively evaluate the quality of motion, such as smoothness and bradykinesia. However, jerk can also be understood as an instantaneous change in force [17]. In particular, it has been reported that the initial contact-LR phase causes a quick impact with the ground, and the ground reaction force is affected by the difference in gait velocity [23]. Therefore, jerk is also lower in patients with slow gait velocity and reduced ground reaction force during the LR phase. Therefore, cases with reduced gliding cannot adapt to force changes in the LR phase. The result is a gait with a reduced change in force in the LR phase.

The jerk of the force change value during gait is also effective in understanding the characteristics of the gait in the IC-LR phases in patients with trochanteric fractures. Additionally, it was found that decreased gliding after trochanteric fracture was not only related to lateral femoral pain but also to loading during gait. Furthermore, our results show that the difficulty in loading during the gait affected the gait speed.

### Study Limitations

The participants of this study were patients with trochanteric fractures and a gait speed of less than 1 m/s. Therefore, the results for participants with a gait speed of more than 1 m/s are not clear. Due to the cross-sectional nature of the study, the causal relationship between gliding and jerk or gait speed is not understood. In the future, it will be necessary to clarify the relationship of gliding between tissues, jerk, and gait ability through intervention studies.

## 5. Conclusions

The jerk of the force change value during gait is effective in understanding the characteristics of gait in the IC-LR phases among patients with trochanteric fractures. The importance of focusing on jerk RMS and IC-LR jerk were indicated for the improvement of the locomotor rehabilitation index. Additionally, gliding is related not only to impairments such as pain, but also to disabilities such as gait. Therefore, gliding has become an important target for physical therapy interventions.

## Figures and Tables

**Figure 1 sensors-22-03842-f001:**
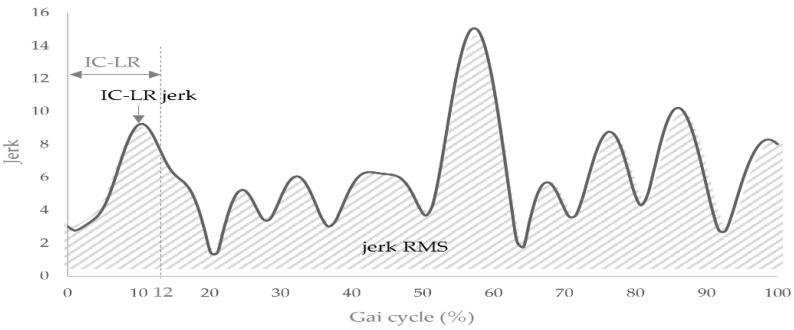
Methods for calculating jerk RMS and IC-LR jerk. Abbreviations: RMS, Root Mean Square; IC-LR, Initial Contact—Loading Response.

**Figure 2 sensors-22-03842-f002:**
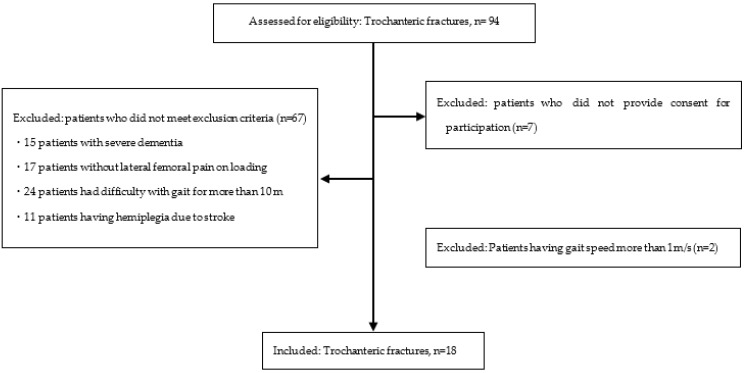
Patient flow scheme accounting for inclusion criteria.

**Figure 3 sensors-22-03842-f003:**
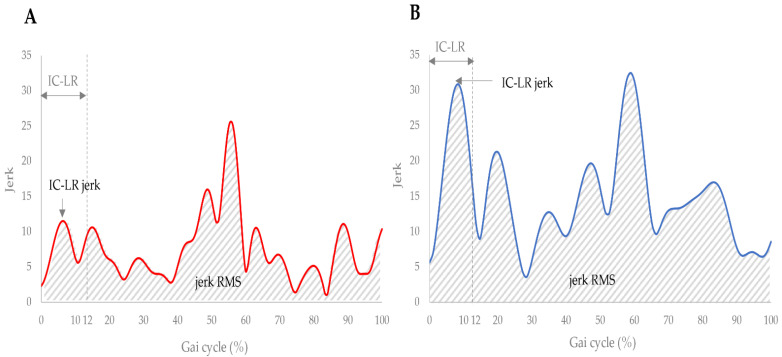
Jerk of excellent and poor cases of gliding between tissues. (**A**) Excellent case, (**B**) Poor case. Abbreviations: RMS, Root Mean Square; IC-LR, Initial Contact—Loading Response.

**Table 1 sensors-22-03842-t001:** Measurement results.

Parameters	Value
Gliding (r)	0.52 ± 0.12
Lateral Femoral Pain	4.2 ± 2.1
Gait Velocity (m/s)	0.56 ± 0.19
Jerk RMS (m/s^3^)	1.4 ± 0.6
IC-LR jerk (m/s^3^)	25.0 ± 12.3
Stride time variability	4.6 ± 3.5
Step time asymmetry	4.5 ± 3.4
Double stance ratio (%)	29.5 ± 5.8
Single stance ratio (%)	38.8 ± 10.0
Locomotor rehabilitation index	40.0 ± 13.6

The following is a list of measurement data. Note: Data are presented as mean SD. Abbreviations: RMS, Root Mean Square; IC-LR, Initial Contact—Loading Response.

**Table 2 sensors-22-03842-t002:** Correlation between gliding and each gait parameter.

	Gliding	
Parameters	r	*p*-Value
Lateral Femoral Pain	0.517	0.016 *
Gait Velocity (m/s)	−0.316	0.163
Jerk RMS (m/s^3^)	−0.433	0.049 *
IC-LR jerk (m/s^3^)	−0.459	0.037 *
Stride time variability	−0.228	0.320
Step time asymmetry	0.202	0.380
Double stance ratio (%)	0.002	0.463
Single stance ratio (%)	0.169	0.993
Locomotor rehabilitation index	−0.341	0.166

Abbreviation: RMS, Root Mean Square; IC-LR, Initial Contact—Loading Response. * *p* < 0.05.

**Table 3 sensors-22-03842-t003:** Correlation between gait speed and each parameter.

	Gait Velocity	
Parameters	r	*p*-Value
Gliding (r)	−0.316	0.163
Lateral Femoral Pain	0.106	0.648
Jerk RMS (m/s^3^)	0.599	0.004 **
IC-LR jerk (m/s^3^)	0.679	<0.001 **
Stride time variability	0.177	0.444
Step time asymmetry	−0.335	0.138
Double stance ratio (%)	−0.608	0.013 *
Single stance ratio (%)	0.531	0.003 **
Locomotor rehabilitation index	0.998	<0.001 **

Abbreviations: RMS, Root Mean Square; IC-LR, Initial Contact—Loading Response. ** *p* < 0.01; * *p* < 0.05.

## Data Availability

The data presented in this study are available on request from the corresponding author.

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
