# Peer review of "Relationship between Tissue Gliding of the Lateral Thigh and Gait Parameters after Trochanteric Fractures"

_sensors, 2022, doi:10.3390/s22103842_

Round 1

Reviewer 1 Report

Main comments

The risk of falls and functional mobility are important current issues in public health. Being able to predict this risk and gait alterations are important goals in the clinical setup. Especially welcome is the use of inertial motion sensors. The study is well structured but needs adjustment on two important topics.

1) The sample calculation and use of sampling. Please indicate that there were 18 subjects evaluated (if I understand correctly) because of the 94 subjects initially rolled into the study only this number reached the inclusion criteria. And please show the sample size calculation in order to understand the power of the tests used.

2) I missed important references on the use of motion sensors that I describe below and suggest the use of a new index that can contribute decisively to your study, addition of the locomotor rehabilitation index, as explained below.

Consider showing in the introduction and discussion, firstly how the biomechanical parameters are impaired in individuals with fractures, using high-level evidence (e.g. PMID: 34351056).

Particularly, I suggest trying to use one relatively new parameter to analyze the functional mobility. First, consider applying the rehabilitation locomotor index. To do this, you need just the walking speed and the lower limb length (great trochanter to the ground) or 0.54 of height (https://www.ncbi.nlm.nih.gov/pmc/articles/PMC2872302/)

After, you need to apply these two simple equations:

OWS (optimal walking speed, in m/s) = sqrt ( 0.25 x 9.81 x lower limb length (or 0.54 of height))

LRI (locomotor rehabilitation index, in %) = 100 x walking speed / OWS

The message is obtain the walking speed normalized based on theory of dynamic similarities and given an index that represents how is the person is close to your more economical metabolically to his/her optimal walking speed and where the pendular mechanism is more optimized. To understand in depth, please read: PMID: 30699211, PMID: 29614469, PMID: 23059867.

Minor points

On this statement:

Lines 87-88 - The STV was calculated as the standard deviation of each stride time divided by the mean stride time.

Consider including a reference supporting the method (e.g.: PMID: 30699211).

Reviewer 2 Report

The authors aim to investigate the relationship between gliding in the VL and subcutaneous tissue and the parameters as measured by inertial sensors and clearly stated the hypothesis and purpose of their study in the last paragraph of the introduction section.

  • line 59 (2.1 Participants): more demographic characteristics should be added.
  • The data collection and data analysis sections are very poor. No procedure is mentioned, (sensors calibration, sensor positioning (why they choose the particular position), protocol of data transmission, initial data collection, and data analysis). please detail.
  • The results presented are poor.  Plots regarding gait parameters behavior are needed, please provide more time trends comparison

The authors mention the purpose of the study and clearly summarize the results in the first paragraph of the discussion section.

The discussion of the results is well presented.

Round 2

Reviewer 2 Report

The authors addressed all issues as recommended.

The manuscript clearly presents the study design, results interpretation and discussion. No more comments for the authors.